# Functional Disability and Brain MRI Volumetry Results among Multiple Sclerosis Patients during 5-Year Follow-Up

**DOI:** 10.3390/medicina59061082

**Published:** 2023-06-04

**Authors:** Sintija Strautmane, Arturs Balodis, Agnete Teivane, Dagnija Grabovska, Edgars Naudins, Daniels Urbanovics, Edgars Fisermans, Janis Mednieks, Alina Flintere-Flinte, Zanda Priede, Andrejs Millers, Maksims Zolovs

**Affiliations:** 1Faculty of Residency, Riga Stradins University, LV-1007 Riga, Latvia; teivanea@gmail.com; 2Department of Neurology, Pauls Stradins Clinical University Hospital, LV-1002 Riga, Latvia; janis.mednieks@gmail.com (J.M.); alina.flintere-flinte@stradini.lv (A.F.-F.); zandapriede@gmail.com (Z.P.); millersandrejs@gmail.com (A.M.); 3Department of Radiology, Riga Stradins University, LV-1002 Riga, Latvia; arturs.balodis7@gmail.com (A.B.); dagnija.grabovska@gmail.com (D.G.); edgars.naudins@gmail.com (E.N.); 4Institute of Diagnostic Radiology, Pauls Stradins Clinical University Hospital, LV-1002 Riga, Latvia; 5Faculty of Medicine, Riga Stradins University, LV-1007 Riga, Latvia; daniel.urbanovics@gmail.com (D.U.); edgars.fisermans@gmail.com (E.F.); 6Department of Neurology and Neurosurgery, Riga Stradins University, LV-1007 Riga, Latvia; 7Statistics Unit, Riga Stradins University, LV-1007 Riga, Latvia; maksims.zolovs@rsu.lv; 8Institute of Life Sciences and Technology, Daugavpils University, LV-5401 Daugavpils, Latvia

**Keywords:** multiple sclerosis, brain atrophy, volumetry, MRI, EDSS, disability

## Abstract

*Background and Objectives*: We aimed to determine the link between brain volumetry results and functional disability calculated using the Expanded Disability Status Scale (EDSS) among multiple sclerosis (MS) patients in relation to the provided treatment (disease-modifying therapies (DMTs)) during a 5-year follow-up period. *Materials and Methods*: A retrospective cohort study was performed enrolling 66 consecutive patients with a confirmed diagnosis of MS, predominantly females (62% (*n* = 41)). Relapsing–remitting (RR) MS was noted in 92% (*n* = 61) of patients, with the rest being patients with secondary progressive (SP) MS. The mean age was 43.3 years (SD 8.3 years). All patients were evaluated clinically using the EDSS and “FreeSurfer© 7.2.0” radiologically during a 5-year follow-up. *Results*: A significant increase in patient functional disability was noted, calculated using the EDSS during a 5-year follow-up. The baseline EDSS ranged between 1 and 6 with a median of 1.5 (IQR 1.5–2.0), and after 5 years, the EDSS was between 1 and 7, with a median EDSS of 3.0 (IQR 2.4–3.6). Compared with RRMS patients, SPMS patients demonstrated a significant increase in EDSS score during a 5-year period, with a median EDSS of 2.5 in RRMS patients (IQR 2.0–3.3) and 7.0 (IQR 5.0–7.0) among SPMS patients. Significantly lower brain volumetry results in different brain areas were found, including cortical, total grey and white matter, *p* < 0.05. Statistically significant differences were observed between baseline volumetry results of the hippocampus and the middle anterior part of the corpus callosum and their volumetry results after 5 years, *p* < 0.001. In this study population, the thalamus did not demonstrate significant changes in volumetry results during follow-up, *p* > 0.05. The provided treatment (DMTs) did not demonstrate a significant impact on the brain MRI volumetry results during a 5-year follow-up, *p* > 0.05. *Conclusions:* Brain MRI volumetry seriously impacts the early detection of brain atrophic changes. In this study, significant relationship between brain magnetic resonance volumetry results and disability progression among MS patients with no important impact of the provided treatment was described. Brain MRI volumetry may aid in the identification of early disease progression among MS patients, as well as enrich the clinical evaluation of MS patients in clinical patient care.

## 1. Introduction

Multiple sclerosis (MS) is the most common immune-mediated disease associated with chronic inflammation and demyelination of the central nervous system (CNS) [1]. The estimated prevalence of MS patients is around 2.8 million people worldwide [2]. Although the exact cause of MS remains unclear, it is known that MS is a heterogeneous disorder characterized by variable clinical and pathological features with inflammation, demyelination, and axonal degeneration as the major mechanisms that cause clinical manifestations.

A recent study on the mechanistic underpinning of an inside-out concept for autoimmunity in MS patients was performed in the Netherlands and published in 2021, suggesting that in the brains of MS patients, there is a disintegration of axon–myelin units, potentially causing the excess systematic release of post-translationally modified myelin. This study supports a core pathogenic role of T cells present in the normal brain that hyper-reacts to post-translationally modified (citrullinated) myelin-oligodendrocyte glycoprotein and evoke clinical and pathological aspects of MS. Data published in this study support the Wilkin’s primary lesion theory: autoimmunity is a physiological response of the immune system against excess antigen turnover in diseased tissue (in the primary lesion) and individuals who develop autoimmune disease are genetically predisposed high responders against critical antigens [3]. Nevertheless, in this study, such mechanisms contributing to MS were not analysed.

In recent years, brain atrophy measurement using magnetic resonance imaging (MRI) has been proposed as a promising marker of tissue damage and neurodegeneration among MS patients. It has been confirmed that measuring the percentage of brain tissue loss over time is one of the best methods of quantifying neurodegeneration and MS monitoring [4]. It is known that patients with MS develop whole-brain atrophy over time, but there are some subcortical structures whose volumetry changes may be used as biomarkers in predicting MS disease progression and an increase in functional disability. For instance, significant atrophy of the putamen, corpus callosum, and caudate nucleus has been proved previously in MS patients, with the thalamus noted as an especially susceptible subcortical structure in the early stages of MS [5,6]. 

A recent study on the progression of regional grey matter atrophy in MS patients, published in 2018, explained that grey matter atrophy was present from the earliest stages of MS. This study aimed to determine the sequence in which grey matter regions become atrophic in MS and its association with disability accumulation. This study enrolled more than 1400 MS patients with different MS phenotypes—clinically isolated syndrome (CIS), RRMS, SPMS and primary progressive MS (PPMS), as well as more than 200 control subjects [7]. Subcortical structures such as the cerebellum, caudate nucleus and putamen demonstrated early atrophy in RRMS and late atrophy in PPMS. This study also mentions the link between atrophy and T2 lesions: T2 lesion load and disease duration in all patients in their study group were associated with increased event-based model stage with no effects on DMTs and comorbidity on event-based model stage. Brain volumes in other diseases such as dementia and Parkinson’s decrease quickly, particularly in the grey matter [7]. This study concluded that over time, grey matter atrophy spreads to involve more regions of the brain, but the sequence in which regions become atrophic is consistent across MS phenotypes. The study suggested that the spread of atrophy was associated with the duration and disability of the disease progression over time in RRMS patients [7].

This study enrolled two MS phenotypes, RRMS and SPMS, predominantly RRMS patients. The patient cohort group in this study was a lot smaller, and this study did not include a control group. This study aimed to determine links between brain volumetry results and functional disability, calculated using EDSS, among MS patients in relation to the provided treatment (DMTs) during a 5-year follow-up. Some subcortical structures demonstrated significantly lower brain MRI volumetry results after a 5-year follow-up, but an analysis of early and late atrophy in MS patients was not studied.

In most studies, brain atrophy is measured using MRI volumetry, and the results are compared between two consecutive points. Grey matter atrophy in MS is thought to be associated with disability and cognitive impairment, and the association between brain atrophy and the increase in MS patient functional disability calculated using the Expanded Disability Status Scale (EDSS) over time has been reported. Nevertheless, only a few publications are available regarding long-term changes in brain volume and their correlation with MS patients’ functional disability concerning the provided treatment. In this study, patient whole brain atrophy and different subcortical structure atrophy concerning functional disability progression and the provided treatment during a 5-year follow-up period were analysed. Patients’ cognitive impairment over time was not studied in this research. The results from this study provide further information on brain MRI volumetry results in MS patients in the long term.

## 2. Materials and Methods

### 2.1. Patient Demographics

In a retrospective cohort study, a total of 66 consecutive patients with a confirmed diagnosis of MS according to the revised McDonald’s criteria were enrolled. This study population included patients with both relapsing–remitting (RR) and secondary progressive (SP) multiple sclerosis (MS), where most patients were RRMS patients, 92% (*n* = 61). The degree of disability was assessed using the Expanded Disability Status Scale (EDSS). EDSS ranges from 0 (normal neurological status) to 10 (death from MS) [8]. Exclusion criteria were patients with other CNS diseases, suboptimal magnetic resonance images, and artefacts in magnetic resonance images that could influence the results of the brain volume analysis.

### 2.2. MRI Acquisition and Analysis

Brain magnetic resonance imaging examinations were performed with a 1.5 T scanner. Images were made in the coronal, axial, and sagittal planes. High-resolution three-dimensional (3D) sequences are preferred for measuring brain volume because they provide great T1-weighted (T1W) image contrast between grey and white matter and show anatomy details [9]. Brain volumetric analysis was performed with an automatic segmentation method using the brain imaging software package FreeSurfer 7.2.0., Harvard, United States. All patients underwent a complete neurological examination at the time of MRI and after a 5-year follow-up. 

### 2.3. Statistical Analysis

The assumption of normal data distribution was assessed using the Shapiro–Wilk test and visual inspection of their histograms and normal Q–Q plots. The assumption of homogeneity of variances was tested using Levene’s test. The presence of outliers was inspected according to Hoaglin (1986, 1987) [10,11]. The two-sided, Kruskal–Wallis H test with pairwise comparisons applying the Bonferroni adjustment was used to determine whether there were statistically significant differences in EDSS between receiving medication (the 1st, 2nd and both lines). The two-sided, Mann–Whitney U test was used to compare EDSS between forms of the disease (relapsing–remitting and secondary progressive). The two-sided, dependent samples *t*-test or Wilcoxon signed-rank test was used to compare brain volumetry results in different brain areas at the beginning of the study and after a 5-year follow-up. Differences were considered statistically significant at *p* < 0.05. Measurements of the strength and direction of the association between age and brain volumetry results were evaluated using the two-sided, Spearman rank order correlation. Correlations were considered statistically significant at *p* < 0.05. These tests were conducted using Jamovi version 2.3 [12].

## 3. Results

### 3.1. Patient Characteristics

In this study group, a total of 66 consecutive patients with a confirmed diagnosis of MS were enrolled, predominantly females, 62% (*n* = 41). The mean age of patients was 43.3 years ranging between 25 and 59 years (SD: 8.3). The study population included patients with both relapsing–remitting (RR) and secondary progressive (SP) multiple sclerosis (MS), where most patients were RRMS patients, 92% (*n* = 61). At the baseline, the patient functional disability score, calculated using the Expanded Disability Status Scale (EDSS), ranged between 1 and 6 with a median score of 1.5 (IQR 1.5–2.0). During a 5-year follow-up, patient disability significantly increased, with EDSS ranging between 1 and 7, with a median score of 3.0 (IQR 2.4–3.6), *p* < 0.001. The increase in functional disability was noted in most patients of the study group, 86% (*n* = 47). Compared with RRMS patients, SPMS patients demonstrated a significant increase in EDSS during a 5-year period, with a median EDSS of 2.5 in RRMS patients (IQR 2.0–3.3) and 7.0 (IQR 5.0–7.0) among SPMS patients, respectively. A comparison of brain atrophy between patients who received treatment and who did not was not performed. 

In this study, first-line medications included interferons: interferon beta 1a (Avonex and Rebif), peginterferon beta 1a (Plegridy), glatiramer acetate (Copaxone), teriflunomide (Aubagio) and dimethyl fumarate (Tecfidera). Avonex was prescribed for six patients in this study group, all RRMS patients. On the other hand, eight RRMS and one SPMS patients received Rebif. Betaferon was prescribed for one patient in this study (SPMS patients). Furthermore, six patients received interferon beta-1b (Extavia), where five of them were SPMS patients. All patients who received Plegridy were RRMS patients, *n* = 9. Copaxone was prescribed for 24 patients—22 patients of them were RRMS patients. Almost all patients who received Tecfidera (*n* = 34) were RRMS patients, *n* = 33. Only three patients received Aubagio, and two of them were RRMS patients. 

Second-line medications included natalizumab (Tysabri), fingolimod (Gilenya), siponimod (Mayzent), ocrelizumab (Ocrevus), cladribine (Mavenclad), ponesimod (Ponvory), ofatumumab (Kesimpta) and mitoxantrone (Novantrone). All patients who received Tysabri (*n* = 7) and Gilenya (*n* = 5) were RRMS patients. On the other hand, twelve patients received Mayzent, predominantly SPMS patients, *n* = 10. RRMS patients received all other second-line medications: Mavenclad and Ocrevus were received by two patients, five patients received Kesimpta and Oncotrone, and Ponvory was prescribed for one patient. 

First-line medications were received by 66% (*n* = 24) of patients. There were 8% (*n* = 5) of patients who only received second-line medication and 26% (*n* = 17) patients who received both, first- and second-line therapy, *p* < 0.001. DMTs in this study did not demonstrate a significant impact on brain MRI volumetry results, *p* > 0.05. 

During a 5-year follow-up, all patient functional disabilities increased. A significant increase in EDSS among patients receiving both first-and second-line medication was noted, *p* < 0.05 (see Table 1).

### 3.2. Brain Volumes

Significantly lower brain volumetry results in different brain areas were discovered, including cortical and total grey and white matter, *p* < 0.05. During 5-year follow-up, this study showed significant differences between changes in volumetry results in total cortical grey matter (*p* < 0.001) and total grey matter (*p* < 0.001), as well as between volumetry results in total cerebral white matter (*p* < 0.001) (see Figure 1). 

In this study, a statistically significant difference between baseline volumetry results of the left and the right hippocampus, and their volumetry results was found, as well as the middle anterior part of corpus callosum after a 5-year follow-up, *p* < 0.001 and *p* = 0.049, respectively. In this study population, the thalamus volumetry changes, along with the central and anterior parts of the corpus callosum did not demonstrate significant change during a 5-year follow-up period, *p* > 0.05 (see Figure 2 and Figure 3). Moreover, the older were patients in our study group, the lower was their total cortical grey matter volume (see the correlation graph in Figure 4).

### 3.3. Expanded Disability Status Scale and Patient Treatment

At baseline, the patient functional disability score, calculated using the Expanded Disability Status Scale (EDSS), ranged between 1 and 6, with a median score of 1.5 (IQR 1.5–2.0). During a 5-year follow-up, the increase in functional disability was noted in most patients of this study group, 86% (*n* = 47), with EDSS ranging between 1 and 7, and a median score of 3.0 (IQR 2.4–3.6), *p* < 0.001. 

At baseline, most RRMS patients only demonstrated a minor functional impairment, whereas SPMS patients seemed to have more limited daily activity. During a 5-year period, a slight but statistically significant increase in RRMS patient functional activity was noted, while SPMS patients demonstrated a significantly higher increase in EDSS, indicating a more rapid development of functional disability while still being able to carry out some daily tasks, with a median EDSS of 2.5 in RRMS patients (IQR 2.0–3.3) and 7.0 (IQR 5.0–7.0) among SPMS patients, respectively. As noted before, nearly all patients in this study group were treated with a specific disease-modifying therapy, 97% (*n* = 64). Nevertheless, in this study group, the provided treatment did not demonstrate a significant impact on patient functional disability or brain MRI volumetry results during a 5-year period, *p* > 0.05 (see Table 2).

## 4. Discussion

According to the available literature data, it is known that MS is driven by autoimmune pathology in CNS, but the trigger of the autoimmune pathogenic process remains unclear. The results of the present work suggest brain MRI volumetry as an important tool for early detection of brain atrophic changes among MS patients, rather than providing a new light on the underlying molecular mechanisms of MS. 

In this study, brain MRI volumetry results concerning patient functional disability, calculated using EDSS, and the provided treatment during a 5-year follow-up were analysed. Patients aged between 18 and 60 years with a confirmed diagnosis of relapsing–remitting (RR) MS and secondary progressive (SP) MS were included. According to the literature, RRMS is by far the most common phenotype of MS, and in this study, most patients were RRMS patients [12]. This study aimed to determine links between brain volumetry results and functional disability, calculated using EDSS, among MS patients in relation to the provided treatment (DMTs) during a 5-year follow-up.

At baseline, the included SPMS patients demonstrated a generally higher level of functional disability, calculated using EDSS, compared with the included RRMS patients. Although SPMS patients had a significantly higher functional disability, they were still able to accomplish most daily activities. During a 5-year follow-up, a significant increase in EDSS was discovered in all patients of this study, especially among SPMS patients. The patient’s cognitive impairment over time was not analysed in this study.

During a 5-year period, significantly lower brain volumetry results in different brain areas, including cortical and total grey and white matter, were found. In this study population, thalamus volumetry changes, along with the central and anterior parts of the corpus callosum did not demonstrate significant differences during a 5-year follow-up period, *p* > 0.05. Previous studies have reported thalamic atrophy to be associated with the progression of disability or cognitive impairment in MS patients [13]. Numerous studies have also proved corpus callosum and deep grey matter as promising biomarkers for MS progression [14]. Some studies have also studied the relation between brain atrophic changes and disability or cognitive impairment, sometimes without significant changes, noting the importance of careful clinical evaluation in MS patients with brain atrophy [6]. 

In this study group, volume loss in the hippocampus was statistically significant. According to the available literature data, relatively little is known of the extent of hippocampal involvement in MS [15]. The volume loss detected in the left and the right hippocampus could mirror demyelination, active inflammatory process, neuronal loss, or other neuropathological processes. Most of these study group patients included RRMS patients who developed various amounts of exacerbations during the 5-year follow-up period, frequently treated with high-dose corticosteroids (methylprednisolone). It is known that exposure to such therapy is another important variable that could have an impact on hippocampal volume [16]. Not only had external treatment been noted to have an impact on hippocampal volume but also high levels of endogenous cortisol have in fact been associated with a negative impact on the hippocampal volume [17]. Although cortisol levels were not a target measure among this study population, they could be informative for future studies. 

On the other hand, a study of lifespan neurodegeneration among multiple sclerosis (MS) was performed in France, published in March 2023, which analysed a much larger patient group for a longer period, and there was also a control group [18]. They found that the most significant changes first occurred in the thalamus, followed by the diencephalon, including the hippocampus. 

When comparing the study performed in France on lifespan neurodegeneration among MS patients with this study, a significant decrease in volumetry results was noted not only in the hippocampus but also in other brain structures, including the middle anterior part of the corpus callosum, total grey matter, and total white matter. This study did not have a control group. One of the possible explanations for the results of this study includes the fact that a lifespan neurodegeneration analysis among MS patients was not performed, as well as limitations in this study—a low number of patients, predominantly RRMS patients. It is possible that if such analysis was performed on a larger patient group, similar results could be obtained. 

In this study, significant volumetry changes in the thalamus during a 5-year follow-up were not found, although some studies report the thalamus as an especially susceptible structure to atrophy at the early stages of MS, along with atrophy in the caudate nucleus, putamen, and corpus callosum [19,20]. 

A study on the progression of regional grey matter atrophy in MS patients, published in 2018, explained that grey matter atrophy was present from the earliest stages of MS [7]. This study concluded that over time, grey matter atrophy spreads to involve more regions in the brain [7]. However, an analysis of early and late atrophy in MS patients in this study was not performed.

As referred before in the Introduction, an article on the mechanistic underpinning of an inside-out concept for autoimmunity in MS patients was performed in the Netherlands and published in 2021, suggesting that in the brains of MS patients, there is a disintegration of axon–myelin units potentially causing the excess systematic release of post-translationally modified myelin. In this study, such mechanisms contributing to MS were not analysed, suggesting the necessity for further studies on this topic. 

Studies have shown that brain atrophy occurs faster in MS patients compared with healthy individuals. MS patients have a smaller brain grey matter volume compared with healthy individuals. Each year, MS patients lose brain tissue faster (0.5–1% per year) than healthy individuals (0.1–0.3% per year). Overall brain atrophy is seen in all subtypes of MS. Brain atrophy can be observed in all stages of the disease and develops gradually. Loss of brain tissue is primarily caused by loss of myelin, but changes in the water content of brain tissue, loss of glial cells, and vascular elements and changes in grey matter result in a decrease in total brain volume. Neurodegeneration in macroscopically visible brain atrophy can be quantified in vivo via magnetic resonance imaging [21].

Recent studies have shown that axonal loss and dysfunction occur early in the disease and continue throughout the disease. Cerebral atrophy is seen not only in white matter foci but also in normal-appearing cerebral bath matter, possibly secondary to changes in myelin loss and axonal damage from Valerian-type degeneration. Many studies report that there is no direct correlation between total brain volume and a patient’s clinical condition, so there are factors that influence total brain volume regardless of disease. The literature mentions that any inflammatory response can cause a temporary increase in total brain volume due to vasogenic oedema, and glial cell proliferation and gliosis can also cause a false increase in brain volume. Biological factors also play an important role; body fluid status, diet, menstrual cycle, genetic and environmental factors, gender, and age can affect total brain volume. It is known that the reduction in total brain volume progresses with age, but other risk factors such as alcohol, smoking, dehydration, malnutrition, protein loss, and other diseases (cardiovascular, Cushing syndrome, etc.) also affect the accuracy of brain volume measurements [21].

It should also be noted that the correlation between brain tissue loss and disease progression is influenced by a phenomenon called brain plasticity. MS patients are thought to have a compensatory mechanism that relies on the activation of new areas of the brain to allow certain brain functions. In the early stage of the disease, functional changes develop in the cerebral cortex, the role of which is to reduce the clinical impact of MS foci damage, because of which many MS patients remain undiagnosed for a long time, especially emphasizing the importance and necessity of studies on correlations between MR volumetry and clinical manifestations of the disease. The accuracy of MRI volumetry measurements is also affected by the phenomenon of pseudoatrophy, which is a transient loss of fluid and a reduction in oedema after initiating anti-inflammatory therapy (a transient decrease in total brain volume during the first 6 months to 1 year after initiating therapy) and is not associated with neuronal or tissue damage [21].

The study limitations include a small sample size, where most patients were RRMS patients with a very low number of SPMS patients. There was no direct control group. If the proportion of RRMS and SPMS patients was more equal, more detailed results about patient functional disability progression and brain volumetry results could be obtained. Grey matter atrophy in MS is thought to be associated with disability and cognitive impairment. Patients’ cognitive impairment over time was not studied in this research. According to the available literature data, there is no direct correlation between the total brain volume and patients’ clinical condition; therefore, it should be kept in mind that there are factors that affect total brain volume regardless of the disease. Moreover, any inflammatory reaction may cause a transitional increase in total brain volume due to vasogenic oedema, as well as the proliferation of glial cells that can result in a false increase in the brain volume. Status of body fluids, menstrual cycle, genetic and environmental factors, gender, and age may also affect total brain volume. While the reduction in the total brain volume is progressive with age, other risk factors such as alcohol, smoking, dehydration, malnutrition, protein loss, and other diseases can affect the accuracy of the brain volume measurement [12]. Almost all patients were treated with DMTs with potentially differential effects on lesion dynamics. First-line medications were received by 66% (*n* = 24) of patients; 8% (*n* = 5) of patients only received second-line medication; and 26% (*n* = 17) of patients received both first- and second-line therapy, *p* < 0.001. Nevertheless, provided treatment (DMTs) did not demonstrate a significant impact on brain MRI volumetry results. This patient cohort demonstrates the heterogeneity of patients in clinical practice.

This study suggests that the hippocampus and middle anterior parts of corpus callosum atrophy could be among the most favourable biomarkers for predicting MS patient disease progression and functional disability, along with thalamus, caudate nucleus, corpus callosum, and putamen atrophy, as reported in other studies.

This is a pilot study in Latvia on brain MRI volumetry results among MS patients. The practical point of this study is to increase knowledge about brain MRI volumetry in MS and its importance, as well as to promote brain MRI volumetry use in clinical practice as early as possible—it may aid in the identification of early disease progression, as well as enrich clinical evaluation of MS patients in clinical patient care. Nevertheless, further research on the topic including patient cognitive testing is mandatory to provide the best care for MS patients. 

## 5. Conclusions

Brain MRI volumetry seriously impacts the early detection of brain atrophic changes. 

The findings of this study indicated a significant relationship between brain magnetic resonance volumetry results and disability progression among MS patients with no important impact of the provided treatment. A significant decrease in brain MRI volumetry results was noted in multiple brain locations, including total cortical grey matter, total grey matter, and total white matter, and subcortical structures such as the hippocampus and partly corpus callosum during a 5-year follow-up. Brain MRI volumetry may aid in the identification of early disease progression among MS patients, as well as enrich the clinical evaluation of MS patients in clinical patient care. 

## Figures and Tables

**Figure 1 medicina-59-01082-f001:**
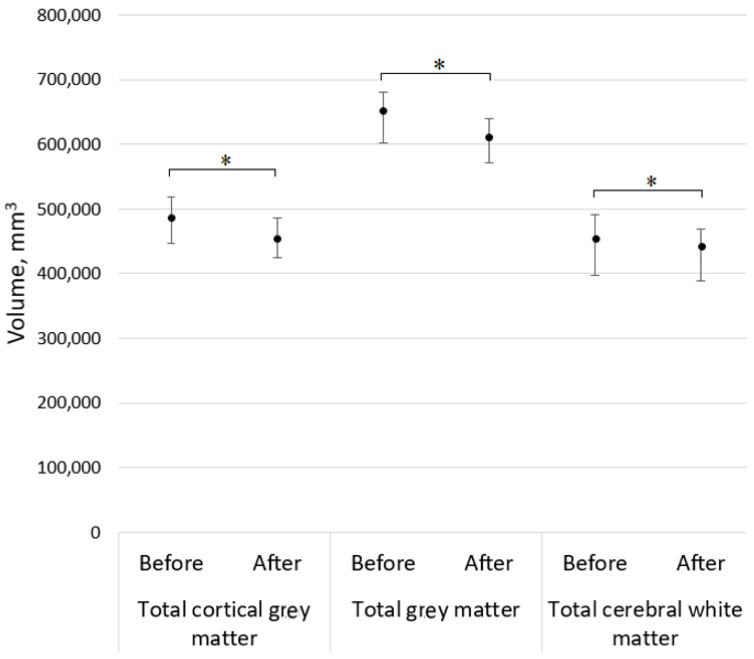
Median with an interquartile range of total cortical grey matter, total grey matter, and total cerebral white matter volumetry results, where asterisks demonstrate a significant difference (*p* < 0.05).

**Figure 2 medicina-59-01082-f002:**
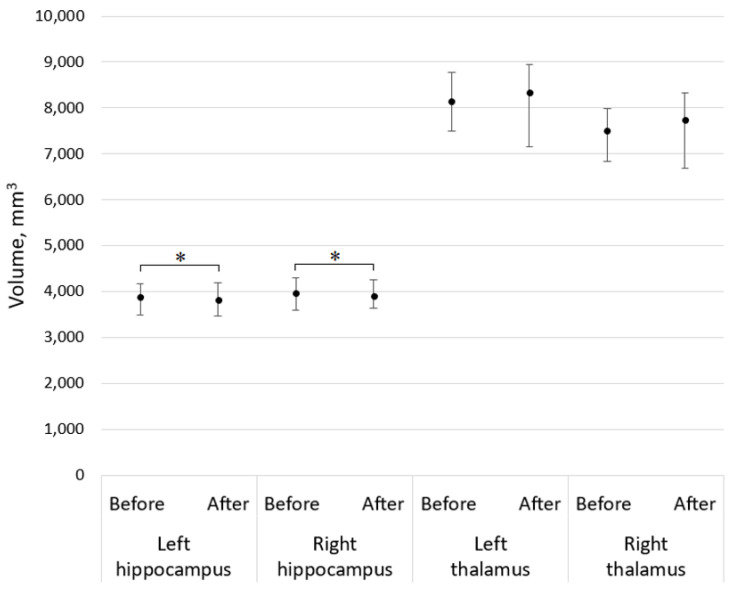
Median with an interquartile range of the left and the right hippocampus and the left and right thalamus volumetry results, where asterisks demonstrate a significant difference (*p* < 0.05).

**Figure 3 medicina-59-01082-f003:**
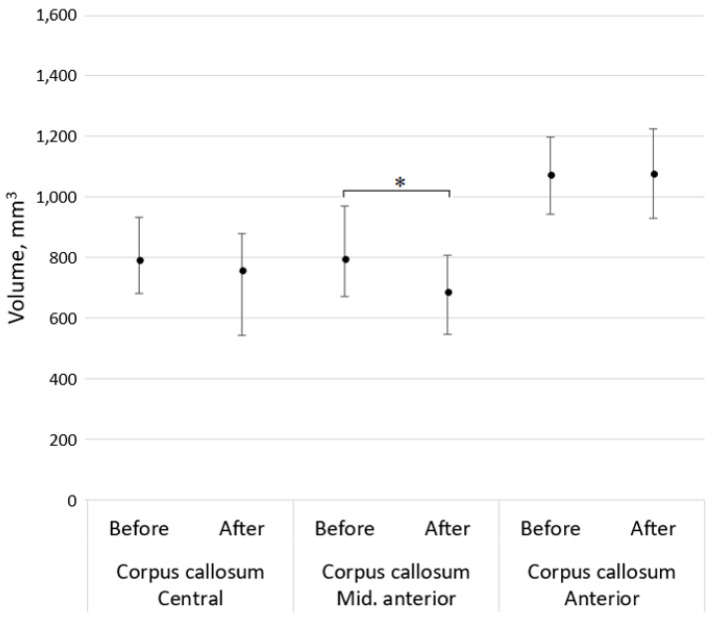
Median with an interquartile range of the central, middle anterior and anterior parts of corpus callosum volumetry results, where asterisks demonstrate a significant difference (*p* < 0.05).

**Figure 4 medicina-59-01082-f004:**
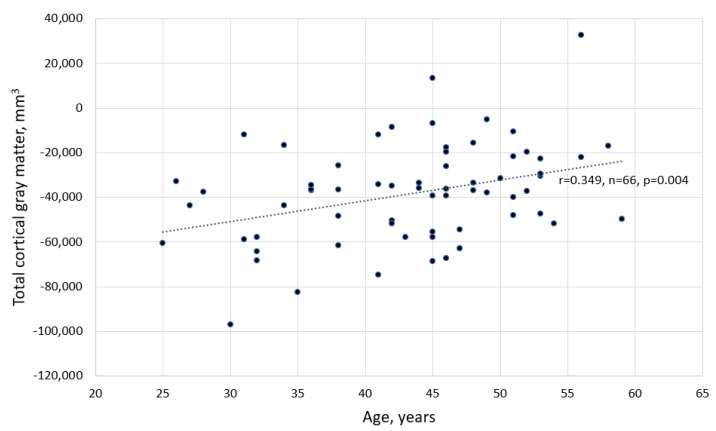
The correlation between differences (before and after) of total cortical grey matter and age.

**Table 1 medicina-59-01082-t001:** Gender, provided treatment and MS phenotype among patients in this study group. SD—standard deviation.

Variables	Count (%) or Mean (SD)	*p*-Value
**Gender**		0.049
Male	38% (*n* = 25)	
Female	62% (*n* = 41)	
**Provided treatment**		<0.001
1st-line therapy	66% (*n* = 42)	
2nd-line therapy	8% (*n* = 5)	
1st- and 2nd-line therapy	26% (*n* = 17)	
**MS phenotype**		<0.001
Relapsing–remitting (RR) MS	92% (*n* = 61)	
Secondary progressive (SP) MS	62% (*n* = 5)	
**Patient age**	43.3 (8.3)	

**Table 2 medicina-59-01082-t002:** The volume of different brain structures among this study population at the baseline and 5-year follow-up. Note: Q1—Q3 is interquartile range presented as the first and third quartiles, r—effect size, NA—not applicable.

Variable	Baseline Volume, mm^3^	Volume at 5-Year Follow-Up, mm^3^	*p*-Value	r
	Median (Q1–Q3)	Median (Q1–Q3)		
Left Hippocampus	3887 (3595–4277)	3816 (3445–4157)	0.001	0.46
Right Hippocampus	3960 (3615–4316)	3902 (3549–4158)	<0.001	0.46
Left Thalamus	8138 (7497–8773)	8229 (7673–9370)	0.061	NA
Right Thalamus	7552 (7173–8309)	7605 (7063–8575)	0.095	NA
Corpus callosum Central	791 (649–901)	759 (641–974)	0.908	NA
Corpus callosum Mid. Anterior	794 (618–916)	687 (569–827)	0.049	0.28
Corpus callosum Anterior	1075 (953–1207)	1077 (930–1223)	0.841	NA
Total cortical grey matter	487,653 (457,323–527,996)	455,258 (424,223–485,430)	<0.001	0.97
Total grey matter	652,330 (62,351–702,969)	610,719 (582,437–650,096)	<0.001	0.99
Total cerebral white matter	454,936 (419,412–513,217)	441,682 (413,982–494,747)	<0.001	0.67

## Data Availability

The data presented in this study are available from the corresponding author upon request.

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
