# Peer review of "Functional Disability and Brain MRI Volumetry Results among Multiple Sclerosis Patients during 5-Year Follow-Up"

_medicina, 2023, doi:10.3390/medicina59061082_

Round 1
Reviewer 1 Report
Major concern:
1. Lack of control group: this paper does not mention a control group, which would help to establish whether the observed changes in brain volumetry and functional disability are specific to MS patients or also occur in the general population. Additionally, without a control group, it is challenging to assess the significance of differences between baseline volumetry results and those obtained after a 5-year follow-up.
2. Limited brain regions studied: The study focuses on specific brain regions (hippocampus, middle anterior part of the corpus callosum, and thalamus), but it would be beneficial to investigate other brain regions involved in MS pathology and their potential relationship with functional disability.
3. No exploration of potential confounding factors: The study does not mention whether the study accounted for potential confounding factors, such as age, disease duration, lifestyle factors, or other comorbidities that could influence brain volumetry results and functional disability. It only shown a correlation between age and differences of total cortical grey matter but no further discussion in the main text.
The paper is well-writtened
Author Response
Point 1: Lack of control group: this paper does not mention a control group, which would help to establish whether the observed changes in brain volumetry and functional disability are specific to MS patients or also occur in the general population. Additionally, without a control group, it is challenging to assess the significance of differences between baseline volumetry results and those obtained after a 5-year follow-up.
Response 1: Thank you for your comment! In this study, we did not have a control group. We agree with you that control group would help to establish whether the observed changes in brain volumetry and functional disability are specific to MS patients or also occur in the general population. This is a pilot study in Latvia on brain MRI volumetry results among MS patients in 5-year period, therefore, further research is warranted.
Point 2: Limited brain regions studied: The study focuses on specific brain regions (hippocampus, middle anterior part of the corpus callosum, and thalamus), but it would be beneficial to investigate other brain regions involved in MS pathology and their potential relationship with functional disability.
Response 2: Thank you for your thoughts! In this study, we analyzed more brain regions, such as caudal anterior part of gyrus cinguli, cuneus, entorhinal cortex, medial part of orbitofrontal cortex, middle temporal, parahippocampal, pericalcarinal cortex, as well as precuneus, rostral anterior part of gyrus cinguli, insula and other regions, but, as there were no significant differences, we did not perform further analysis. We focused more on those structures that demonstrated statistically significant differences (hippocampus, total cortical grey matter, total grey matter, total cerebral white matter and middle anterior part of the corpus callosum).
Point 3: No exploration of potential confounding factors: The study does not mention whether the study accounted for potential confounding factors, such as age, disease duration, lifestyle factors, or other comorbidities that could influence brain volumetry results and functional disability. It only shown a correlation between age and differences of total cortical grey matter but no further discussion in the main text.
Response 3: Thank you for this comment! We demonstrated a correlation between age and differences of total cortical grey matter, and in a paragraph where we explain limitations of this study, we mentioned that status of body fluids, menstrual cycle, genetic and environmental factors, gender and age may also affect total brain volume, as well as age, other risk factors such as alcohol, smoking, dehydration, malnutrition, protein loss and other diseases can affect the accuracy of the brain volume measurement. In this study, the older patients were, the lower their brain volume was, but we did not analyze these factors in detail.
Reviewer 2 Report
This is a very good publication that adds useful information.
As described in the introduction the cause of MS is still unclear. Therefore my advise is to add a broader background in the introduction, including publications such as: "Hart et al. 2021. Mechanistic underpinning of an inside–out concept for autoimmunity in MS. Annals of Clinical and Translational Neurology 8,8,1709–1719."
Then in the discussion, considerations should be taken on whether or not the results of the present work may through new light on the ongoing discussion on the underlying molecular mechanisms of MS.
Author Response
Point 1: This is a very good publication that adds useful information.
Response 1: Thank you very much for your comment!
Point 2: As described in the introduction the cause of MS is still unclear. Therefore my advise is to add a broader background in the introduction, including publications such as: "Hart et al. 2021. Mechanistic underpinning of an inside–out concept for autoimmunity in MS. Annals of Clinical and Translational Neurology 8,8,1709–1719."
Response 2: Thank you for your advice! I rewrote the introduction including the publication you mentioned.
Point 3: Then in the discussion, considerations should be taken on whether or not the results of the present work may through new light on the ongoing discussion on the underlying molecular mechanisms of MS.
Response 3: Thank you very much! I have added this consideration in the discussion part (see the revised Article).
Reviewer 3 Report
Review
Functional disability and brain MRI volumetry results among multiple sclerosis patients during 5-year follow-up.
This article deals with a very interesting topic: the possible use of brain volumetry as a biomarker of degeneration in people with MS, namely in the total and cortical areas of the grey and white matter, in the hippocampus and in the anterior part of the corpus callosum.
Despite of the presented results, the article has several deficiencies in the experimente design in the scientific contente and in the discussion, that make it unpublishable and in need of major alterations.
Abstract
Line 34- EDSS is mentioned by the first time- should be Expanded Disability Status Scale (EDSS)
Line 34 – Multiple sclerosis should be MS
Line 39- remove SD
Line 35 and 51- “provided treatment” this is not clear! DMTs?
Line 51- “our study group” should be this study group
All the Abstract needs an english editing
1-Introduction
It is very poor and should be improved
Line 59- the reference to MS in Europe is too old (2017)
Line 63- “magnetic resonance (MRI) imaging” should be magnetic resonance imaging (MRI)
Line 72- References 5 and 6- these studies mention a point that could be further explored:
Grey matter atrophy is present from the earliest stages of multiple sclerosis (“not even” as the authors say) – several authors observed this fact and are not mentioned like Eshaghi, A et al, 2018. Also the same authors mention the link between atrophy and T2 lesions, also not explored here. Also, brain volumes in other diseases like Dementia and Parkinson's decrease quickly particularly in the grey matter. This is also not mentioned.
Line 72-77- Regarding functinal disability , Gray matter atrophy in MS is thought to be associated with disability and also cognitive impairment, but there are discordant results that the authors should adress
All the Introduction needs an english editing and the authors, again , should be more carefull and remove “our” ………
2-Material and Methods
It must be rewritten
2.1 Patients
-remove our
-remove gender/age
- correct numbers , ex line 90 – (n=61) and not (n=41)
- refer to EDDS here
- refer to ethical approval and patient consent
2.2 MRI acquisition and analysis
2.3 Statistical analysis
Material and Methods needs an english editing
3-Results
Line 123- SD
Line 126 – n=61
Line 134- disease modifying therapy (DMT)
Line 133-138 and table 1-I don’t know what do you mean by 1st line and 2nd line therapy, this should be explained previously
Figure 1 (it is a legend, should be in the bottom of the Figure not on top )- same comment for the other figures.
Line 209-212- Where are the results supporting that 97% of the patients outcomes where not bias by the DMTs ?
Where is the model used to explain/ or not, the association between DMTs and atrophy progression in MS? Is there any comparison, with and without treatment. Treatments with fingolimod and alemtuzumab have been shown to reduce the progression of brain atrophy while most other DMTs have shown only minimal or controversial results
In Table 2:
Thalamic volume and Corpus callosum have non significant results (except CC anterior)- it should be discussed next . Previous authors observed the opposite, ex Marja Niiranen, 2022
4-Discussion
Overall it is very poor and does not have a logical sequence. The english also contributes to the “confusion”.
Lines 221-2237 224 and 225- remove spacing in RRMS and SPMS
Line 229- “as expected” – reference
Line 238 – “disability or cognitive impairment”
Brain atrophy appears during the progression of MS and is associated with the disability caused by the disease; Lack of cognitive testing is another subject also not adressed by the authors.
Line 242-243- volume loss in hippocampus
The authors should discuss the normal volume loss occuring due to age, stage of the disease, certain treatments, ex natalizumab (Koskimaki et al., 2018; Miller et al., 2007). resolution of inflammation after therapy initiation, probably due to fluid shifts (i.e., resolution of brain edema) and changes in inflammatory cells and so on.
Line 246- know corticoids are mentioned……
Line 249- “endogenous cortisol” is mentioned but there are no back up data/results
Line 253- Why mention this study alone? Specially when the experimente design id not comparable at all?!
Line 268- Study limitations
This is a big part of the manuscript: in 66 patients only 61 had RRMS and 5 SPMS- the sample is too short and the number of patients with SPMS even more
This study has several limitations, including a small sample size (66 subjects) with no direct control group ; only 5 patients with SPMS. Also almost all the patients was treated with DMTs, with potentially differential effects on lesion dynamics. Furthermore it is not described the treatments and if the patients changed- first to second line treatments or vice-versa.
References
To check all, eg ref. 11 no date (?)
__________________________________________/__________________________________
All the Manuscript needs an extensive english editing
Round 2
Reviewer 1 Report
All of my concerns have been addressed.
Author Response
Thank you very much for your comments and reply!
Reviewer 3 Report
The authors adressed evey comment and made the requested corrections. Furthermore they improved the scientific content of the introduction and the discussion. From my point of view there are only small mistakes to correct.
Line 144- “central nervous system” should be CNS
Line 192 – “Redif” should be Rebif
Line 196- “Extavia” should be interferon beta-1b (Extavia)
Line 203 -“mitoxantrone (Oncotrone)” should be mitoxantrone (Novantrone)
Line 220- Table 1 title should be in the top not in bottom
Line 298- “central nervous system” should be CNS
Line 367 – This paragraph is identical to the one in introduction (line 71). Maybe the authors could say “ As referred before in the introduction, …………”
___________________________________/______________________________________
